# Effect of Agitation on the Dissolution of APAM with Different Molecular Weights and the Equivalent Diameter of Coal Slime Settling Floc with Different Particle Sizes

Jianbo Li *, Wei Zhou *, Chuanchuan Cai, Shujie Wang and Jinbo Zhu

State Key Laboratory of Mining Response and Disaster Prevention and Control in Deep Coal Mines, College of Materials Science and Engineering, Anhui University of Science and Technology, Huainan 232001, China
* Correspondence: desperato1203@163.com (J.L.); awaly@126.com (W.Z.)

**Abstract:** There are still many problems in the stirring dissolution of flocculants: The law of stirring dissolution of flocculants with different molecular weights is not clear, the microstructure of settling floc with different particle sizes of coal slime is still to be explored, and how to apply the law of flocculant dissolution to industrialization? To solve these problems, firstly, mechanical stirring dissolution tests are carried out on anionic polyacrylamide (APAM) with different molecular weights to explore the stirring dissolution law of APAM with 3–25 million molecular weights. Results indicated that the optimal stirring speed of APAM with 3–6 million molecular weight is 800 r/min. The optimal stirring speed of 8–16 million molecular weight APAM is 600 r/min. The optimal stirring speed of 18–25 million molecular weight APAM is 800 r/min. The stirring power formula for APAM dissolution is improved by fitting the experimental data of 3–25 million molecular weight APAM stirring dissolution. Secondly, through APAM performance test, it is verified that the solubility of powdered APAM is the best under optimal solution. Thirdly, through the image measurement and analysis test of settling floc of coal slime, the change law of equivalent diameter of settling floc of different particle sizes of coal slime under the optimal dissolution parameters are explored. Results showed that when the mixing speed is 600 r/min, the equivalent diameter of −0.5 mm slime settling floc is the largest. When the mixing speed is 200 r/min or 400 r/min, the equivalent diameters of 0.5–0.25 mm, 0.25–0.125 mm, 0.125–0.075 mm, 0.075–0.045 mm and −0.045 mm slime settling floc are the largest. The equivalent diameter is positively correlated with the size of slime, and the equivalent diameter of −0.045 mm slime settling floc is the largest. The size of slime is the dominant factor in the change of fractal dimension. The fractal dimension of −0.5 mm slime settling floc is the largest, and the fractal dimension is little affected by mixing speed. Finally, the relationship between the important dissolution parameters of the APAM stirring dissolution test is numerically fitted. In view of the different geometrical parameters of the stirring barrel, a method of deducing the dissolution and stirring time of industrial flocculants is proposed.

**Keywords:** APAM; molecular weight; stirring speed; stirring power; equivalent diameter; fractal dimension; stirring time

## 1. Introduction

The effects of anionic polyacrylamide (APAM) on slurry are as follows: (i) enhanced the aggregation between particles; (ii) promoted the gradual transformation of active free water into stable adsorbed water and floc water; (iii) made the particles more closely arranged through adsorption bridging [1]. Applying an appropriate dosage of APAM has a positive effect on kaolinite aggregation and subsequent settling [2,3]. Shear force is the main force of flocculant stirring dissolution. The influence of shear force on aggregation dynamics depends on particle size [4]. Reflocculation of clay suspensions formed by uncharged or low-charged polymers is not obvious after the initial floc are broken under

high shear forces. In contrast, high charge-density polymers do not form large initial floc, but they show significant reflocculation ability under a continuous shear condition [5].

Most polymer chains cannot be released until the maximum gel size is reached, depending on the initial size of the flocculant. The aging time of flocculant preparation increases with the increase of particle size. The decrease of coarse particle size leads to the decrease of the optimum aging time of flocculant solution [6]. For powdered products, the initial wetting, swelling, and dispersion of the solid polymer may be affected by the powder size, and the polymer moves from discrete chains to highly entangled agglomerates containing many chains in solution, reaching maximum activity in 48 to 72 h. Dissolution of powdered polyacrylamide flocculants is a slow process, and the dosage required to achieve measurable flocculation decreases as the flocculant age increases to 72 h, re-reflecting the release of discrete polymer chains from the highly agglomerated solution species. Although flocculant consumption is much higher after shorter aging periods, aging time has no significant effect on the density of the formed aggregates, suggesting that only the discrete polymer chains affect the flocculation process [7,8]. The settling rate of the flocculating slurry depends on the polymer concentration and the method of flocculant addition. The sedimentation rate and the clarity of the supernatant depend on the mixing intensity. The sedimentation rate decreases with the increase of turbulence, but the clarity shows an obvious maximum. The deterioration of flocculant activity over time occurred mainly in the first seven days and increased at elevated temperatures [9].

Using flocculant solution as the front end of coal slime water treatment is a necessary condition to accelerate the settling of coal slime water. In recent years, the production practice has proved that anionic polyacrylamide (APAM) is better than cationic polyacrylamide (CPAM) and nonionic polyacrylamide (NPAM) in the treatment of coal slime water, and the molecular weight of APAM is more widely used, so APAM is selected as the research object. On the basis of previous studies, the stirring dissolution law of APAM with different molecular weights was discussed, and the optimal dissolution performance of solid APAM was verified under the optimal dissolution condition. Then, the changes of equivalent diameter of coal slime settling floc with different particle sizes were studied. Finally, an industrial flocculant solution—stirring time derivation method was proposed.

## 2. Test

### 2.1. Test Materials and Instruments

(1) APAM Stirring Dissolution Test

The test subject was APAM: 3 million, 6 million, 8 million, 10 million, 12 million, 16 million, 18 million, 20 million, 22 million, 25 million molecular weight (Henan Zhongbang Environmental Protection Technology Co., Ltd., Zhengzhou, China). As shown in Figure 1, the main instruments included 76-1A glass thermostatic water bath, NP-40LS cantilever electric stirrer, NDJ-97 viscometer and DDS-11C conductivity meter.

(2) APAM Performance Test

The test material mainly included 10 kg magnetic separation tailings from Pan Yi coal preparation plant, and 18 million molecular weight APAM. The main test instruments included Smartlab SE X-ray diffractometer, 5E-SSB200 standard sieve machine, KER-1 muffle furnace and 752N visible spectrophotometer.

(3) Equivalent Diameter Test of Slime Settling Floc

The test materials mainly included −0.5 mm, 0.5–0.25 mm, 0.25–0.125 mm, 0.125–0.075 mm, 0.075–0.045 mm and −0.045 mm particle size slime and 800 r/min configuration of 18 million molecular weight APAM. The main test instrument included SMART1010 image measuring instrument system (Shenzhen Serein Metrology Co., Ltd., Shenzhen, China).

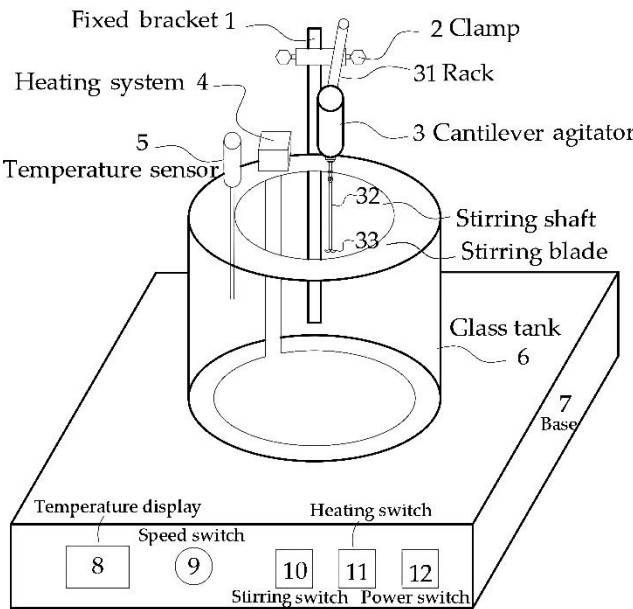

**Figure 1.** Glass thermostatic water bath.

*2.2. Test Methods and Significance*

1. APAM Stirring Dissolution Test

Weigh 10 L of water with a measuring cylinder and pour it into the 76-1A glass thermostatic water bath (shown in Figure 1), set the stirring speed of NP-40LS cantilever electric stirrer to 200–1600 r/min, each group of 200 r/min increments, a total of 8 groups, corresponding to the blade linear speed of 0.21–1.68 m/s; Set the thermostatic temperature to 25 °C, wait for the temperature to rise to the set value, and 5 g of powdered APAM weighed by the electronic balance was poured gently and evenly along the edge of the vortex generated by stirring, and the APAM solution of 0.5‰ concentration was configured, and stirring was continued until the conductivity value of the conductivity meter stopped when there was no obvious change in 3 min; The NDJ-97 viscometer was set to rotor 0 and the rotational speed was 60 r/min, and the APAM solution was measured and recorded. The kinetic viscosity of the dissolved APAM solution was measured and recorded; The density of the dissolved solution was measured and calculated using the weighing method; Formulas (1)–(13) were used to calculate the impeller linear speed, kinematic viscosity of the dissolved solution, stirring power per unit volume, and input energy per unit volume.

The molecular weight and stirring speed were used as test variables for APAM stirring dissolution test. The optimal agitation speed corresponding to the maximum kinematic viscosity of APAM with the same molecular weight was selected to explore the stirring dissolution law of APAM with different molecular weights, improve the calculation formula of agitation power, and provide the optimal reagent dissolution parameters for the subsequent APAM flocculation performance measurement test.

2. APAM Performance Test

The analytic factor test design was used to discriminate the performance of APAM. The concentration of coal slime water was 100 g/L, and APAM solution was added from 0.2 mL to 2.4 mL, mixed with slime water, increasing by 0.2 mL each time, a total of 12 groups, until the settlement effect was rapidly stratified. The settlement rate and degree of clarification were used as test evaluation indexes to select the stirring speed corresponding to the optimal performance of flocculant solution. After standing and settling for 30 min, the supernatant was poured into the spectrophotometer by decanting method, and the maximum value of clarity was recorded. The distilled water was the standard liquid, and the clarity was 100%. In order to determine the mineral composition, the Smartlab SE type X-ray diffractometer was used to analyze the coal samples from Pan Yi coal preparation plant. The diffraction

conditions were as follows: Cu/Kα target, tube voltage of 40 kV, tube current of 30 mA, and diffraction angle of 3–70°.

The APAM performance test was conducted to verify whether the powdered APAM had the best solubility under the optimal stirring speed selected in the APAM stirring dissolution test.

3. Equivalent Diameter Test of Slime Settling Floc

(1) Coal slime particle size classification: Vibrating screen screening −0.5 mm, 0.5–0.25 mm, 0.25–0.125 mm, 0.125–0.075 mm, 0.075–0.045 mm, −0.045 mm as coal samples.

(2) Flocculant solution was configured: 18 million APAM solution with 0.5‰ concentration was configured in glass constant temperature water bath according to the optimal stirring speed N = 800 r/min and stirring time T = 65.23 min selected by APAM stirring dissolution test (According to Table 1).

**Table 1.** Selection of stirring dissolution test results for APAM with different molecular weights.

| Molecular Weight /(Million) | Stirring Speed $N$/(r/min) | Linear Velocity $v$/(m/s) | Stirring Time $T$/(min) | Kinematic Viscosity $\gamma$/($\times 10^{-3}$m$^2$/s) | Reynolds Number $Re$ | Power Number $K$ | Stirring Power per Unit Volume $P_v$/(kW/m$^3$) | Energy Input per Unit Volume $E_v$/($\times 10^3$kJ/m$^3$) |
|---|---|---|---|---|---|---|---|---|
| 3 | 800 | 0.84 | 65.50 | 1.32 | 12.73 | 2.57 | 1.94 | 7.62 |
| 6 | 800 | 0.84 | 64.27 | 1.53 | 10.98 | 2.74 | 2.06 | 7.94 |
| 8 | 600 | 0.63 | 65.28 | 1.69 | 7.46 | 3.35 | 1.06 | 4.15 |
| 10 | 600 | 0.63 | 68.63 | 1.63 | 7.73 | 3.29 | 1.04 | 4.28 |
| 12 | 600 | 0.63 | 68.27 | 1.63 | 7.73 | 3.29 | 1.05 | 4.30 |
| 16 | 600 | 0.63 | 69.48 | 2.15 | 5.86 | 3.87 | 1.22 | 5.09 |
| 18 | 800 | 0.84 | 65.23 | 2.34 | 7.18 | 3.43 | 2.58 | 10.10 |
| 20 | 800 | 0.84 | 65.82 | 2.04 | 8.24 | 3.17 | 2.39 | 9.44 |
| 22 | 800 | 0.84 | 66.63 | 1.93 | 8.70 | 3.08 | 2.31 | 9.23 |
| 25 | 800 | 0.84 | 67.25 | 2.30 | 7.30 | 3.39 | 2.54 | 10.25 |

(3) Mixing of slime and flocculant: The target size slime and water were mixed with 20 g/L slime water in 500 mL beaker. After full mixing, 0.32 mL APAM solution was added (According to Table 2). The NP-40LS cantilever electric agitator was used to mechanically stir the slime water containing APAM for 1 min and 15 min. The mixing speed was set at 200–1000 r/min, increasing every 200 r/min, and another group was added as the control group for verification. In the control group, there were 6 groups in total, with neither flocculant nor mechanical stirring.

**Table 2.** Optimal dissolution parameters range of APAM with different molecular weights.

| APAM Molecular Weight /(Million) | Optimal Stirring Speed /(r/min) | Optimal Linear Velocity /(m/s) | Optimal Stirring Time /(min) | Optimal Kinematic Viscosity /($\times 10^{-3}$m$^2$/s) | Stirring Power per Unit Volume /(kW/m$^3$) | Energy Input per Unit Volume /($\times 10^3$kJ/m$^3$) |
|---|---|---|---|---|---|---|
| 3–6 | 800 | 0.84 | 64–66 | 1.3–1.6 | 1.9–2.1 | 7.6–8.0 |
| 8–16 | 600 | 0.63 | 65–70 | 1.6–2.2 | 1.0–1.3 | 4.1–5.1 |
| 18–25 | 800 | 0.84 | 65–68 | 1.9–2.4 | 2.3–2.6 | 9.2–10.3 |

(4) Photographing floc particles of coal slime: The TopPette pipette with a diameter of 5 mm at the nozzle was used to sample the settling floc in the compressed layer after the flocculation settlement of the coal slime water, and the floc was dropped on the slide. The image measurement system was used to enlarge the floc particles on the slide by 300 times for photographing. In the sampling process, multi-point sampling was needed, and 4 clear images with small error were processed and then the average value was taken.

(5) Image processing: Matlab and Adobe Photoshop CS5 were used to carry out gray scale and binarization processing for each image, and image-Pro-Plus 6.0 was used to

analyze the characteristic parameters of statistical floc: floc area, floc perimeter, aspect ratio, mesopore area, statistical diameter, where the scale is 10 μm.

With the mixing speed as the test variable and the equivalent diameter as the evaluation standard, the optimal mixing speed of the target coal slime and flocculant was selected.

### 2.3. Data Calculation

Where $v$ is the linear velocity of the impeller, m/s; $d$ is the impeller diameter, m; $N$ is the number of revolutions of the impeller per minute, r/min; $\gamma$ is the kinematic viscosity of solution after dissolution, m²/s; $\mu$ is the dynamic viscosity of the solution after dissolution, Pa.s; $\rho$ is the density of solution after dissolution, kg/m³; $Re$ is the Reynolds number.

$$v = \pi d N \tag{1}$$

$$\gamma = \frac{\mu}{\rho} \tag{2}$$

$$Re = \frac{\rho v d}{\mu} = \frac{v d}{\gamma} \tag{3}$$

$$P = K\rho N^3 d^5 \tag{4}$$

$$P_v = \frac{P}{V} \tag{5}$$

$$E_v = \frac{PT}{V} = P_v T \tag{6}$$

where $P$ is the stirring power, W; $K$ is the power number; $P_v$ is the stirring power per unit volume, W/m³; $V$ is the solution volume, m³; $E_v$ is the input energy per unit volume of solution, J/m³; $T$ is the stirring time, s.

$$K = \frac{A^*}{Re} + B^* \left( \frac{10^3 + 1.2Re^{0.66}}{10^3 + 3.2Re^{0.66}} \right)^{C^*} \times \left( \frac{H}{D} \right)^{(0.35 + \frac{u}{D})} (\sin \beta)^{1.2} \tag{7}$$

$$A^* = 14 + \left( \frac{u}{D} \right) \left[ 670 \left( \frac{d}{D} - 0.6 \right)^2 + 185 \right] \tag{8}$$

$$B^* = 10^{[1.3 - 4(\frac{u}{D} - 0.5)^2 - 1.14\frac{d}{D}]} \tag{9}$$

$$C^* = 1.1 + 4\frac{u}{D} - 2.5 \left( \frac{d}{D} - 0.5 \right)^2 - 7 \left( \frac{u}{D} \right)^4 \tag{10}$$

where $K$ is the power number, which is calculated by Nagata power number method [10]; $H$ is the liquid level height, m; $D$ is the diameter of stirring barrel, m; $u$ is the impeller width, m; $\beta$ is the impeller Angle, °; $A^*$, $B^*$ and $C^*$ are intermediate variables.

$$P = f(\rho, d, D, N) = K\rho^a d^b D^c N^e \tag{11}$$

$$\frac{ML^2}{T^3} = K \left( \frac{M}{L^3} \right)^a L^b L^c \left( \frac{1}{T} \right)^e \tag{12}$$

$$P = K\rho N^3 \left( \frac{d}{D} \right)^b D^c \tag{13}$$

where $M$ is the mass dimension; $L$ is the length dimension; $T$ is the time dimension.

$$d^* = \frac{c^* \times a^*}{100} \times \frac{10^6}{b^*} \tag{14}$$

where $d*$ is the solid dosage of flocculant required to treat 1 tonne of slime water, g/t; $a*$ is the concentration of flocculant, %; $b*$ is the concentration of slime water, g/L; $c*$ is the volume dosage of flocculant required to treat 1 L slime water, mL/L.

$$\Phi = 2\sqrt{\frac{S}{\pi}} \times \left[1 - \left(1 - \frac{2\sqrt{S\pi}}{C}\right)k_1\right] \times \left[1 - \left(1 - \frac{1}{n}\right)k_2\right] \times \left(1 - \frac{S_0}{S}k_3\right) \tag{15}$$

$$S = k\Phi^{D_f} \tag{16}$$

$$\ln S = D_f \ln \Phi + \ln k \tag{17}$$

where $\Phi$ is the equivalent diameter of the floc, μm; $S$ is the floc area, μm$^2$; $C$ is the floc perimeter, μm; $n$ is the floc aspect ratio; $S_0$ is the floc mesopore area, μm$^2$; $k_1$, $k_2$ and $k_3$ are the discount factors for perimeter, aspect ratio and mesopore area respectively, which are decimals from 0 to 1, and are taken as 0.5; $k$ is the constant of proportionality; $D_f$ is the mean fractal dimension.

## 3. Results and Analysis

(1) Tables 1 and 2 indicate that the optimum stirring speed $N$ = 800 r/min for 3–6 million molecular weight APAM corresponds to an impeller linear velocity $v$=0.84 m/s; The optimum stirring speed $N$ = 600 r/min for 8–16 million molecular weight APAM corresponds to an impeller linear velocity $v$ = 0.63 m/s; The optimum stirring speed $N$ = 800 r/min for 18–25 million molecular weight APAM corresponds to an impeller linear velocity $v$ = 0.84 m/s.

(2) Figures 2–4 indicate that when the stirring speed ranges from 200 to 1600 r/min, the corresponding impeller linear velocity ranges from 0.21 to 1.68 m/s, the dissolution parameters are positively correlated: kinematic viscosity and molecular weight, Reynolds number and linear velocity, power number and molecular weight, stirring power and linear velocity, stirring power and molecular weight, stirring time and molecular weight; There are negative correlations between Reynolds number and molecular weight, power number and linear velocity, stirring time and linear velocity.

(3) Based on the original stirring power formula, the single factor d of impeller diameter is replaced by the two factors $D$ of stirring bucket diameter and the ratio of the impeller to stirring bucket diameter $d/D$ as the influencing parameters of stirring power. Through data fitting, the improved stirring power formula applicable to the dissolution of APAM is $P = K\rho N^3 D^4 \left(\frac{d}{D}\right)^{5.45}$, which is obtained by fitting all the data from the stirring dissolution test of 3–25 million APAM.

To explore the dissolution law of APAM with different molecular weights, it is necessary to study the relationship between important dissolution parameters, including the relationship between stirring speed and kinematic viscosity, Reynolds number, stirring power per unit volume, power number, and stirring time.

### 3.1. Kinematic Viscosity versus Stirring Speed

Figure 2 indicates that when the stirring speed ranges from 200 to 1600 r/min, the corresponding impeller linear velocity ranges from 0.21 to 1.68 m/s, kinematic viscosity is positively correlated with molecular weight overall, with higher molecular weight increasing kinematic viscosity overall. The kinematic viscosity of the dissolved 3–25 million APAM solution peaks at stirring speeds of $N$ = 600 or 800 r/min ($v$ = 0.63 or 0.84 m/s), with the 16 and 18 million APAM solutions having the highest viscosity and the APAM solution having the lowest viscosity at $N$ = 200 r/min ($v$ = 0.21 m/s).

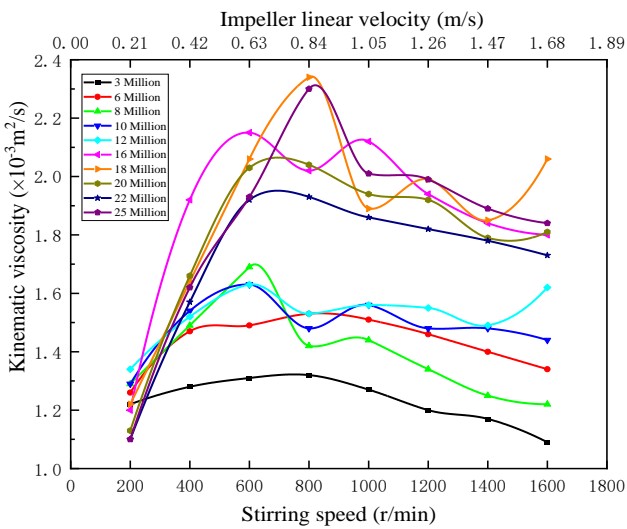

**Figure 2.** Kinematic viscosity versus stirring speed.

### 3.2. Reynolds Number & Stirring Power per Unit Volume versus Stirring Speed

Figure 3 indicate that when the stirring speed ranges from 200 to 1600 r/min, the corresponding impeller linear velocity ranges from 0.21 to 1.68 m/s, both Reynolds number and stirring power per unit volume are positively correlated with the stirring speed. Reynolds number and negatively correlated with the overall molecular weight, under the same stirring speed, low molecular weight polyacrylamide dissolved after the viscosity of the solution is often lower than that of high molecular weight polyacrylamide solution, this is due to the high molecular weight polymer molecular chain segments to entwine, flow resistance get bigger, with the bigger of viscosity and the numerical hours, while Reynolds number effect in fluid force to viscous force, Therefore, Reynolds number of low molecular weight polymers with low viscosity is larger than that of high molecular weight polymers. The stirring power per unit volume is positively correlated with the molecular weight as a whole. Under the same stirring speed, stirring power per unit volume of APAM with high molecular weight is larger than that of APAM with low molecular weight, which is directly related to the solution viscosity. The thicker the solution, the greater the resistance, the more work done.

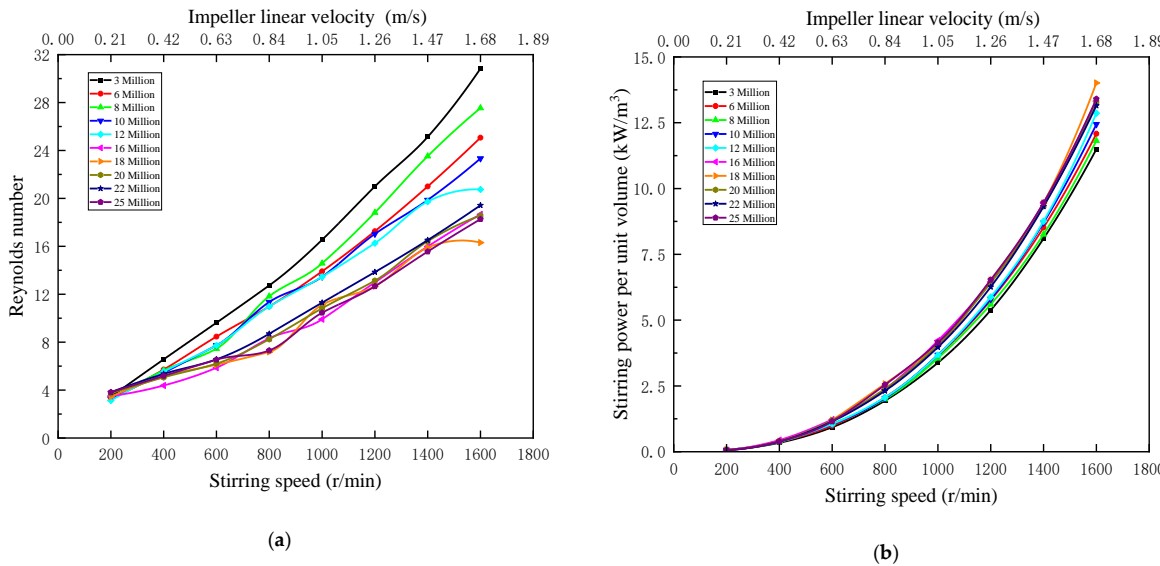

**Figure 3.** (**a**) Reynolds number versus stirring speed; (**b**) Stirring power per unit volume versus stirring speed.

### 3.3. Power Number & Stirring Time versus Stirring Speed

Figure 4 indicate that when the stirring speed ranges from 200 to 1600 r/min, the corresponding impeller linear velocity ranges from 0.21 to 1.68 m/s, both power number and stirring time are negatively correlated with stirring speed. Power number is positively correlated with the molecular weight. At the same stirring speed, power number of low molecular weight APAM is lower than that of high molecular weight APAM. Stirring time is positively correlated with molecular weight as a whole. At the same impeller linear velocity, flocculants with relatively high molecular weight need longer stirring time than flocculants with relatively low molecular weight. Flocculants with high molecular weight have a longer molecular chain and higher viscosity than flocculants with low molecular weight, so the stirring time is longer.

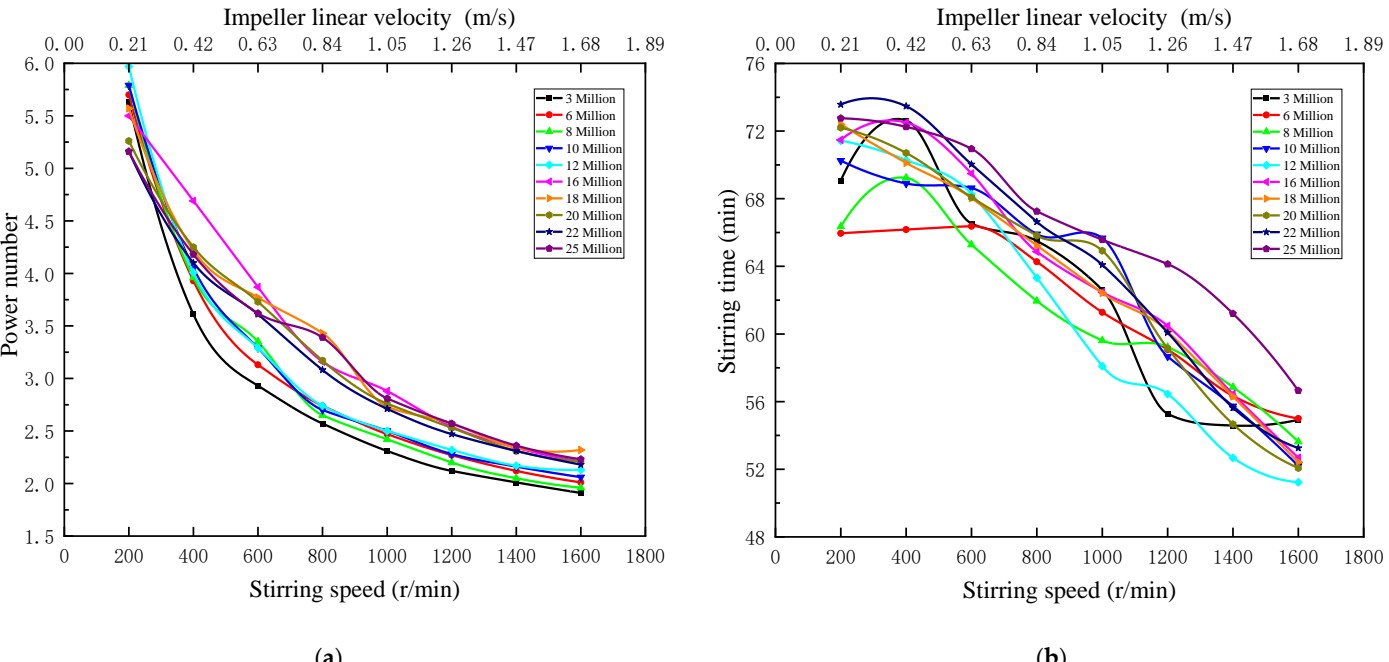

(**a**)　　　　　　　　　　　　　　　　　(**b**)

**Figure 4.** (**a**) Power number versus stirring speed; (**b**) Stirring time versus stirring speed.

### 3.4. APAM Flocculation Performance Determination at Different Stirring Speeds

The analysis results are shown in Figure 5a. The minerals in the magnetic separation tailings of Pan Yi coal preparation plant are mainly quartz ($SiO_2$) and a certain amount of kaolinite ($AlSi_2O_5(OH)_4$). From Table 3 and Figure 5b, when the stirring speed was 800 r/min, the flocculation settlement was the fastest, and the clarification and compressible layer were the highest. The optimal speed of 18 million molecular weight APAM for treating 100 g/L coal slime water is 800 r/min, the optimal volume dosage is 3.2–3.6 mL/L, and the optimal solid dosage is 16–18 g/t (Formula (14)). It verifies that the flocculant solution configured at the optimal stirring speed selected in APAM stirring dissolution test has the optimal flocculation performance. The flocculant solution at the optimal stirring speed selected by APAM with other molecular weights is verified by analogy to have the optimal flocculation performance theoretically.

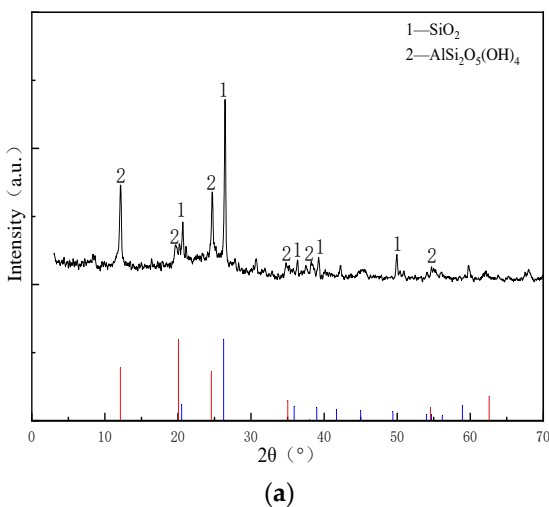

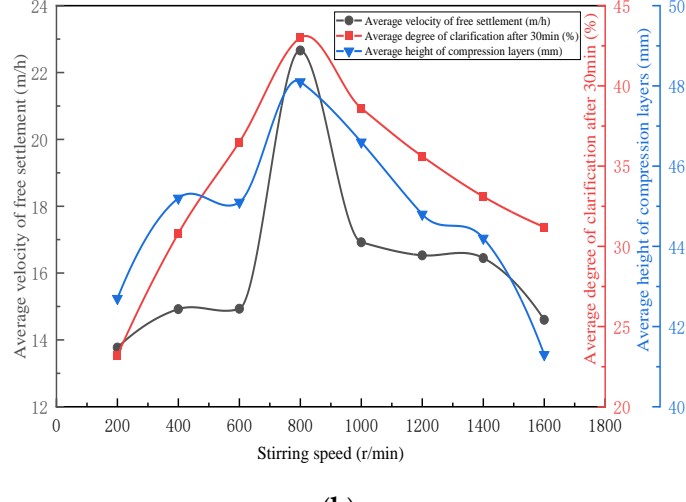

(**a**)　　　　　　　　　　　　　　　　　　(**b**)

**Figure 5.** (**a**) Analysis of mineral composition of coal sample; (**b**) Flocculation determination of main parameters versus stirring speed (18 million molecular weight APAM).

**Table 3.** Performance test results of 18 million molecular weight APAM under optimal solution.

| APAM Stirring Speed /(r/min) | APAM Stirring Time /(min) | Time of Interface Settlement /s | Velocity of Free Settlement /(m/h) | Degree of Clarification after 30 min /% | Height of Compression Layers /mm | APAM Volume Dosage /(mL/L) | APAM Solid Dosage /(g/t) |
|---|---|---|---|---|---|---|---|
| 200 | 72.40 | 32.77 | 13.77 | 23.2 | 42.7 | 3.2–3.6 | 16–18 |
| 400 | 70.12 | 30.02 | 14.92 | 30.8 | 45.2 | 3.2–3.6 | 16–18 |
| 600 | 68.03 | 29.98 | 14.93 | 36.5 | 45.1 | 3.2–4.0 | 16–20 |
| 800 | 65.23 | 19.82 | 22.66 | 43.0 | 48.1 | 3.2–3.6 | 16–18 |
| 1000 | 62.42 | 25.88 | 16.92 | 38.6 | 46.6 | 3.2–4.0 | 16–20 |
| 1200 | 60.15 | 27.49 | 16.53 | 35.6 | 44.8 | 2.8–3.2 | 14–16 |
| 1400 | 56.30 | 28.10 | 16.45 | 33.1 | 44.2 | 2.4–2.8 | 12–14 |
| 1600 | 52.42 | 30.67 | 14.60 | 31.2 | 41.3 | 3.2–3.6 | 16–18 |

## 4. Equivalent Diameter of Settling Floc with Different Particle Sizes of Slime

Through APAM performance test, it was verified that the solubility of powdered APAM was the best under the optimal stirring speed, which provided the optimal dissolution parameters for coal slime water mixing. Therefore, the front work of coal slime water mixing—preparation of flocculant solution was configured according to the optimal dissolution parameters in APAM stirring dissolution test, and then the influence law of coal slime particle size and mixing speed on the equivalent diameter of coal slime settling floc was studied.

Through screening analysis and ash burning test, the proportion and ash content of each size of slime are shown in Table 4. The coal sample includes −0.5 mm, 0.5–0.25 mm, 0.25–0.125 mm, 0.125–0.075 mm, 0.075–0.045 mm and −0.045 mm particle size after screening, among which the maximum proportion of particle size is 0.5–0.25 mm, and the highest ash content is 0.075–0.045 mm.

**Table 4.** Analysis of the percentage of each particle size and ash content of coal samples.

| Slime Size/mm | Percentage of This Level/% | Ash/% |
|---|---|---|
| −0.5 | 100 | 44.30 |
| 0.5–0.25 | 52.17 | 39.20 |
| 0.25–0.125 | 20.03 | 45.26 |
| 0.125–0.075 | 21.45 | 46.93 |

**Table 4.** *Cont.*

| Slime Size/mm | Percentage of This Level/% | Ash/% |
|---|---|---|
| 0.075–0.045 | 6.29 | 49.60 |
| −0.045 | 0.06 | 40.50 |

*4.1. Mixing Speed versus Equivalent Diameter of Settling Floc (−0.5 mm Slime)*

Table 5 is analyzed by Image-Pro-Plus 6.0 software and (Formulas (15)–(17)), which shows that when the mixing speed is 600 r/min, the equivalent diameter of −0.5 mm slime settling floc is the largest, and the settling speed of slime floc is the fastest on the microscopic level. Figure 6a is the analysis diagram of floc area of −0.5 mm slime at the mixing speed of 0–1000 r/min. Figure 6b shows the fractal dimension fitting diagram, when the mixing speed is 0–1000 r/min, the mean fractal dimension of −0.5 mm coal slime floc is 2.142.

**Table 5.** Mixing speed versus equivalent diameter of settling floc (−0.5 mm slime).

| Mixing Speed $N^*$/(r/min) | Floc Area $S$/μm² | Floc Perimeter $C$/μm | Floc Aspect Ratio | Mesopore Area $S_0$/μm² | Equivalent Diameter $\Phi$/μm | Statistical Diameter $D^*$/μm | Floc Fractal Dimension |
|---|---|---|---|---|---|---|---|
| 0 | 13.844 | 13.517 | 1.764 | 0.020 | 3.247 | 2.952 | 2.106 |
| 200 | 332.100 | 100.191 | 1.418 | 0.614 | 14.405 | 19.691 | 2.121 |
| 400 | 487.159 | 117.184 | 1.465 | 0.900 | 17.455 | 23.625 | 2.113 |
| 600 | 621.536 | 147.912 | 1.453 | 1.919 | 18.938 | 27.140 | 2.137 |
| 800 | 250.463 | 97.007 | 1.573 | 1.709 | 11.487 | 16.916 | 2.202 |
| 1000 | 217.873 | 82.418 | 1.523 | 1.219 | 11.246 | 15.504 | 2.164 |

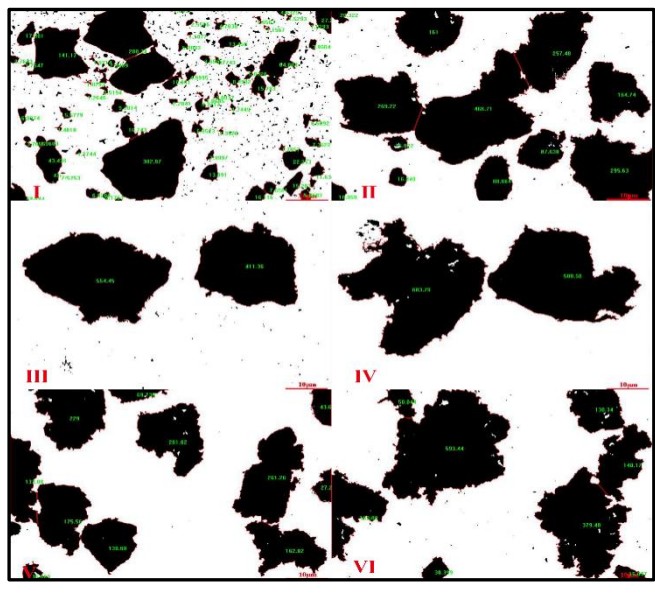

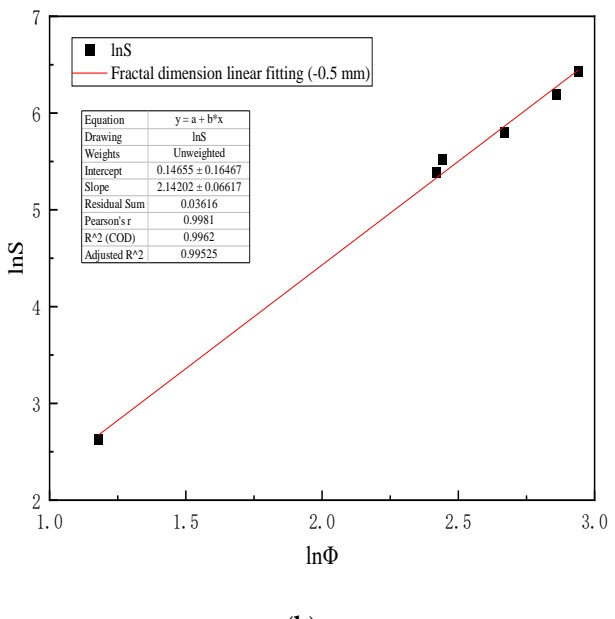

(**a**)　　　　　　　　　　　　　　　　　　　(**b**)

**Figure 6.** (**a**) Area diagram of −0.5 mm coal slime floc at different mixing speeds. (**I**) 0, (**II**) 200 r/min, (**III**) 400 r/min, (**IV**) 600 r/min, (**V**) 800 r/min, (**VI**) 1000 r/min; (**b**) Fractal dimension fitting diagram of settling floc of −0.5 mm coal slime.

*4.2. Mixing Speed versus Equivalent Diameter of Settling Floc (0.5–0.25 mm Slime)*

Table 6 is analyzed by Image-Pro-Plus 6.0 software and (Formulas (15)–(17)), which shows that when the mixing speed is 200 r/min, the equivalent diameter of 0.5–0.25 mm slime settling floc is the largest, and the settling speed of slime floc is the fastest on the

microscopic level. Figure 7a is the analysis diagram of floc area of 0.5–0.25 mm slime at the mixing speed of 0–1000 r/min. Figure 7b shows the fractal dimension fitting diagram, when the mixing speed is 0–1000 r/min, the mean fractal dimension of 0.5–0.25 mm coal slime floc is 1.901.

**Table 6.** Mixing speed versus equivalent diameter of settling floc (0.5–0.25 mm slime).

| Mixing Speed $N^*$/(r/min) | Floc Area $S$/μm² | Floc Perimeter $C$/μm | Floc Aspect Ratio | Mesopore Area $S_0$/μm² | Equivalent Diameter $\Phi$/μm | Statistical Diameter $D^*$/μm | Floc Fractal Dimension |
|---|---|---|---|---|---|---|---|
| 0 | 8.796 | 15.154 | 1.911 | 0.155 | 2.140 | 2.515 | 1.893 |
| 200 | 210.463 | 71.005 | 1.472 | 0.643 | 11.832 | 15.733 | 1.868 |
| 400 | 232.714 | 90.271 | 1.454 | 1.049 | 11.588 | 16.437 | 1.925 |
| 600 | 161.343 | 71.092 | 1.418 | 0.924 | 9.952 | 13.533 | 1.893 |
| 800 | 147.104 | 70.947 | 1.528 | 1.287 | 9.051 | 12.741 | 1.933 |
| 1000 | 133.455 | 57.871 | 1.596 | 0.744 | 9.026 | 11.729 | 1.891 |

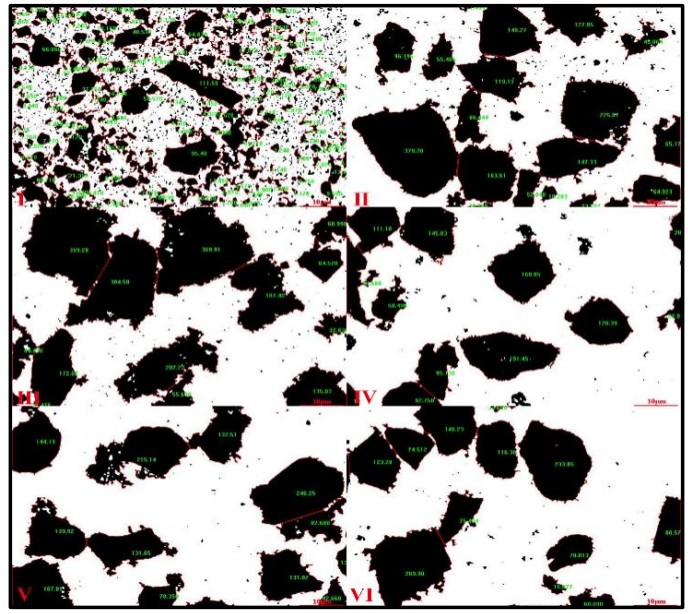

(**a**)

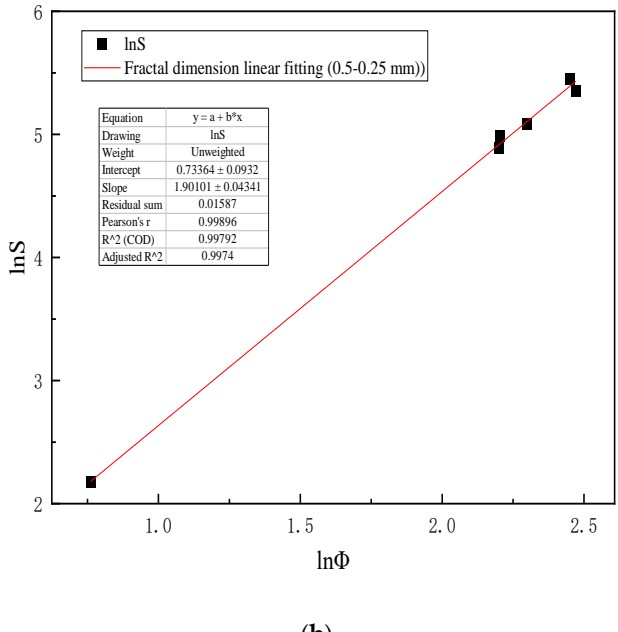

(**b**)

**Figure 7.** (**a**) Area diagram of 0.5–0.25 mm coal slime floc at different mixing speeds. (**I**) 0, (**II**) 200 r/min, (**III**) 400 r/min, (**IV**) 600 r/min, (**V**) 800 r/min, (**VI**) 1000 r/min; (**b**) Fractal dimension fitting diagram of settling floc of 0.5–0.25 mm coal slime.

### 4.3. Mixing Speed versus Equivalent Diameter of Settling Floc (0.25–0.125 mm Slime)

Table 7 is analyzed by Image-Pro-Plus 6.0 software and (Formulas (15)–(17)), which shows that when the mixing speed is 400 r/min, the equivalent diameter of 0.25–0.125 mm slime settling floc is the largest, and the settling speed of slime floc is the fastest on the microscopic level. Figure 8a is the analysis diagram of floc area of 0.25–0.125 mm slime at the mixing speed of 0–1000 r/min. Figure 8b shows the fractal dimension fitting diagram, when the mixing speed is 0–1000 r/min, the mean fractal dimension of 0.25–0.125 mm coal slime floc is 2.016.

**Table 7.** Mixing speed versus equivalent diameter of settling floc (0.25–0.125 mm slime).

| Mixing Speed $N^*$/(r/min) | Floc Area $S$/μm² | Floc Perimeter $C$/μm | Floc Aspect Ratio | Mesopore Area $S_0$/μm² | Equivalent Diameter $\Phi$/μm | Statistical Diameter $D^*$/μm | Floc Fractal Dimension |
|---|---|---|---|---|---|---|---|
| 0 | 8.809 | 15.046 | 1.855 | 0.124 | 2.174 | 2.603 | 2.011 |
| 200 | 318.213 | 117.278 | 1.731 | 1.450 | 12.192 | 18.481 | 2.059 |
| 400 | 526.273 | 134.425 | 1.765 | 1.172 | 16.253 | 24.452 | 2.027 |
| 600 | 347.649 | 104.718 | 1.638 | 0.835 | 13.801 | 19.316 | 1.995 |
| 800 | 319.651 | 97.070 | 1.630 | 0.480 | 13.441 | 18.969 | 1.983 |
| 1000 | 213.312 | 87.115 | 1.663 | 0.698 | 10.501 | 15.223 | 2.019 |

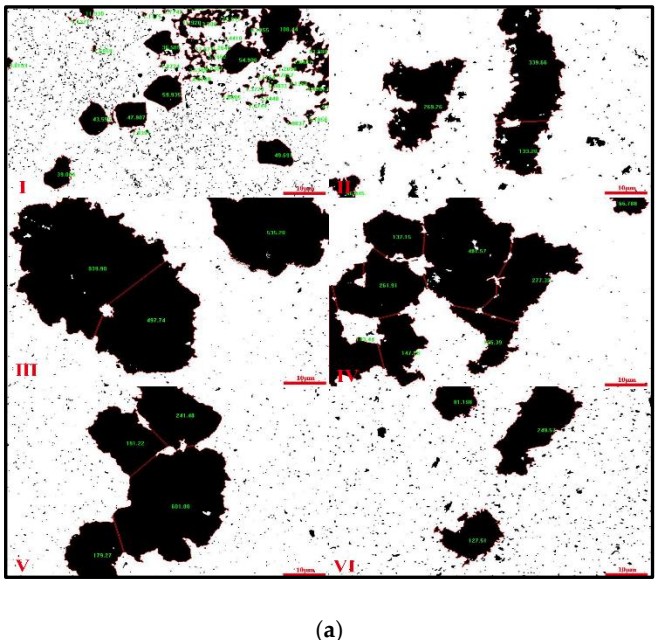

(**a**)

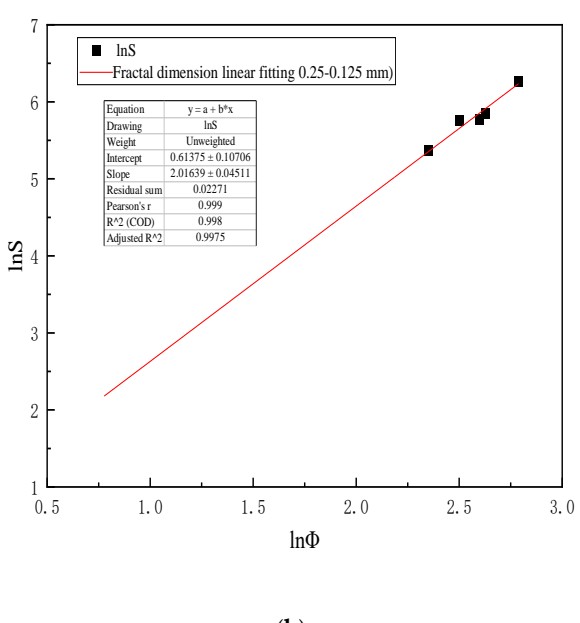

(**b**)

**Figure 8.** (**a**) Area diagram of 0.25–0.125 mm coal slime floc at different mixing speeds. (**I**) 0, (**II**) 200 r/min, (**III**) 400 r/min, (**IV**) 600 r/min, (**V**) 800 r/min, (**VI**) 1000 r/min; (**b**) Fractal dimension fitting diagram of settling floc of 0.25–0.125 mm coal slime.

### *4.4. Mixing Speed versus Equivalent Diameter of Settling Floc (0.125–0.075 mm Slime)*

Table 8 is analyzed by Image-Pro-Plus 6.0 software and (Formulas (15)–(17)), which shows that when the mixing speed is 200 r/min, the equivalent diameter of 0.125–0.075 mm slime settling floc is the largest, and the settling speed of slime floc is the fastest on the microscopic level. Figure 9a is the analysis diagram of floc area of 0.125–0.075 mm slime at the mixing speed of 0–1000 r/min. Figure 9b shows the fractal dimension fitting diagram, when the mixing speed is 0–1000 r/min, the mean fractal dimension of 0.125–0.075 mm coal slime floc is 2.099.

**Table 8.** Mixing speed versus equivalent diameter of settling floc (0.125–0.075 mm slime).

| Mixing Speed $N^*$/(r/min) | Floc Area $S$/μm² | Floc Perimeter $C$/μm | Floc Aspect Ratio | Mesopore Area $S_0$/μm² | Equivalent Diameter $\Phi$/μm | Statistical Diameter $D^*$/μm | Floc Fractal Dimension |
|---|---|---|---|---|---|---|---|
| 0 | 10.155 | 13.978 | 1.784 | 0.019 | 2.534 | 3.151 | 2.068 |
| 200 | 1234.052 | 209.218 | 1.559 | 3.533 | 25.911 | 38.391 | 2.066 |
| 400 | 1488.903 | 294.515 | 1.756 | 9.195 | 24.941 | 41.258 | 2.149 |
| 600 | 516.473 | 129.028 | 1.410 | 3.016 | 17.747 | 24.637 | 2.035 |
| 800 | 412.032 | 127.182 | 1.811 | 2.416 | 13.876 | 21.151 | 2.139 |
| 1000 | 126.904 | 64.890 | 1.710 | 0.788 | 8.110 | 11.986 | 2.125 |

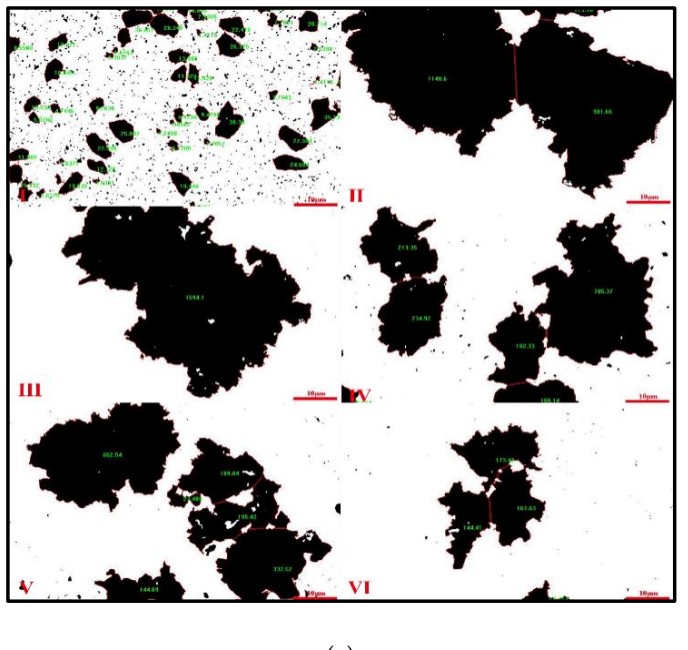

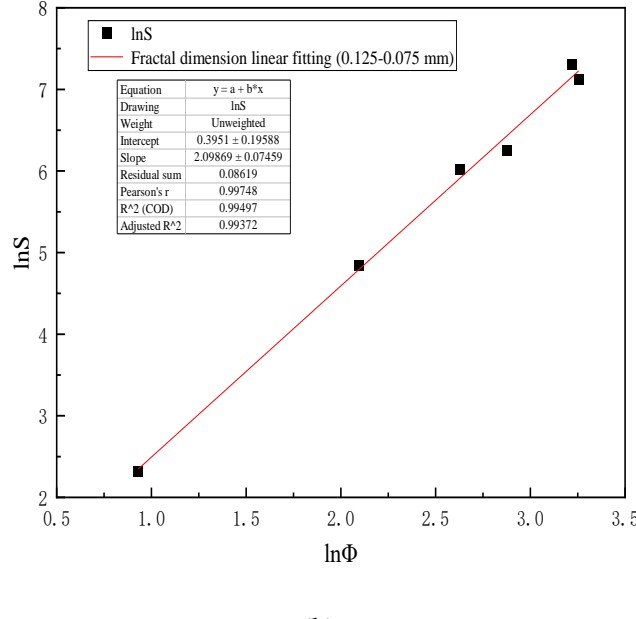

(**a**)

(**b**)

**Figure 9.** (**a**) Area diagram of 0.125–0.075 mm coal slime floc at different mixing speeds. (**I**) 0, (**II**) 200 r/min, (**III**) 400 r/min, (**IV**) 600 r/min, (**V**) 800 r/min, (**VI**) 1000 r/min; (**b**) Fractal dimension fitting diagram of settling floc of 0.125–0.075 mm coal slime.

### 4.5. Mixing Speed versus Equivalent Diameter of Settling Floc (0.075–0.045 mm Slime)

Table 9 is analyzed by Image-Pro-Plus 6.0 software and (Formulas (15)–(17)), which shows that when the mixing speed is 200 r/min, the equivalent diameter of 0.075–0.045 mm slime settling floc is the largest, and the settling speed of slime floc is the fastest on the microscopic level. Figure 10a is the analysis diagram of floc area of 0.075–0.045 mm slime at the mixing speed of 0–1000 r/min. Figure 10b shows the fractal dimension fitting diagram, when the mixing speed is 0–1000 r/min, the mean fractal dimension of 0.075–0.045 mm coal slime floc is 2.073.

**Table 9.** Mixing speed versus equivalent diameter of settling floc (0.075–0.045 mm slime).

| Mixing Speed $N^*$/(r/min) | Floc Area $S$/μm² | Floc Perimeter $C$/μm | Floc Aspect Ratio | Mesopore Area $S_0$/μm² | Equivalent Diameter $\Phi$/μm | Statistical Diameter $D^*$/μm | Floc Fractal Dimension |
|---|---|---|---|---|---|---|---|
| 0 | 3.379 | 8.300 | 1.708 | 0.021 | 1.463 | 1.930 | 2.096 |
| 200 | 1602.686 | 261.753 | 1.660 | 1.841 | 27.892 | 43.195 | 2.091 |
| 400 | 1054.716 | 194.764 | 1.782 | 2.550 | 22.729 | 33.940 | 2.094 |
| 600 | 1008.134 | 187.500 | 1.474 | 2.127 | 24.032 | 34.498 | 2.043 |
| 800 | 481.434 | 133.881 | 1.542 | 2.811 | 16.085 | 23.397 | 2.072 |
| 1000 | 293.505 | 93.803 | 1.612 | 1.128 | 12.876 | 18.047 | 2.059 |

### 4.6. Mixing Speed versus Equivalent Diameter of Settling Floc (−0.045 mm Slime)

Table 10 is analyzed by Image-Pro-Plus 6.0 software and (Formulas (15)–(17)), which shows that when the mixing speed is 400 r/min, the equivalent diameter of −0.045 mm slime settling floc is the largest, and the settling speed of slime floc is the fastest on the microscopic level. Figure 11a is the analysis diagram of floc area of −0.045 mm slime at the mixing speed of 0–1000 r/min. Figure 11b shows the fractal dimension fitting diagram, when the mixing speed is 0–1000 r/min, the mean fractal dimension of −0.045 mm coal slime floc is 1.940.

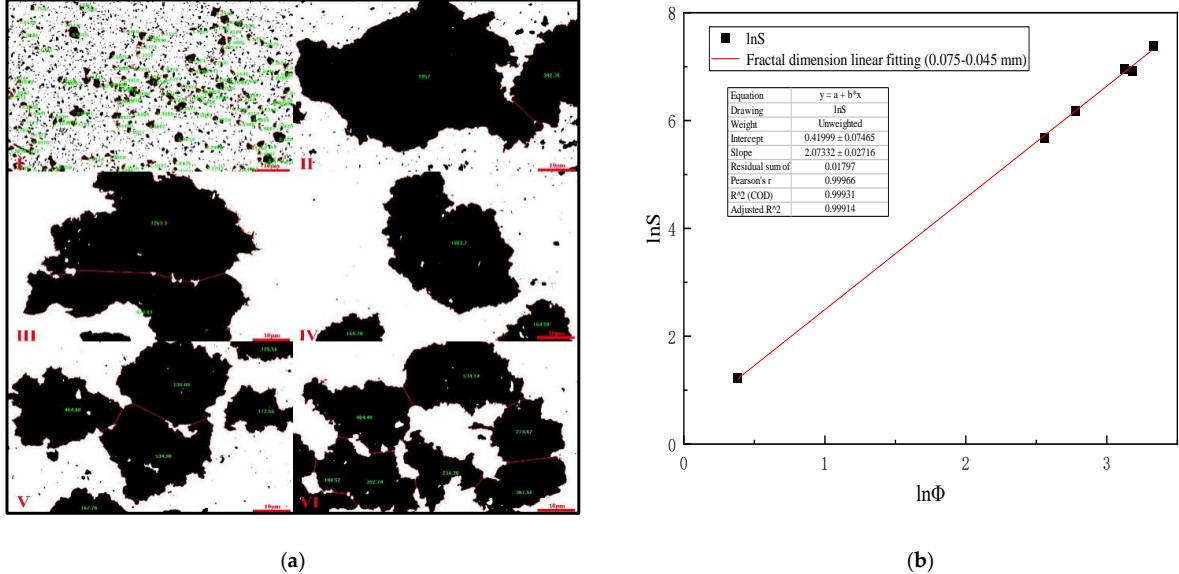

(**a**)　　　(**b**)

**Figure 10.** (**a**) Area diagram of 0.075–0.045 mm coal slime floc at different mixing speeds. (**I**) 0, (**II**) 200 r/min, (**III**) 400 r/min, (**IV**) 600 r/min, (**V**) 800 r/min, (**VI**) 1000 r/min; (**b**) Fractal dimension fitting diagram of settling floc of 0.075–0.045 mm coal slime.

**Table 10.** Mixing speed versus equivalent diameter of settling floc (−0.045 mm slime).

| Mixing Speed $N^*$/(r/min) | Floc Area $S$/μm$^2$ | Floc Perimeter $C$/μm | Floc Aspect Ratio | Mesopore Area $S_0$/μm$^2$ | Equivalent Diameter $\Phi$/μm | Statistical Diameter $D^*$/μm | Floc Fractal Dimension |
|---|---|---|---|---|---|---|---|
| 0 | 2.354 | 8.692 | 1.909 | 0.027 | 1.066 | 1.637 | 1.958 |
| 200 | 1358.856 | 197.174 | 1.670 | 4.624 | 27.597 | 39.726 | 1.954 |
| 400 | 2143.444 | 285.286 | 1.436 | 4.370 | 34.865 | 51.044 | 1.954 |
| 600 | 702.405 | 162.639 | 1.740 | 1.173 | 18.558 | 28.231 | 1.994 |
| 800 | 521.996 | 100.775 | 1.317 | 1.514 | 20.422 | 24.848 | 1.832 |
| 1000 | 450.453 | 132.859 | 1.550 | 2.575 | 15.384 | 23.013 | 1.968 |

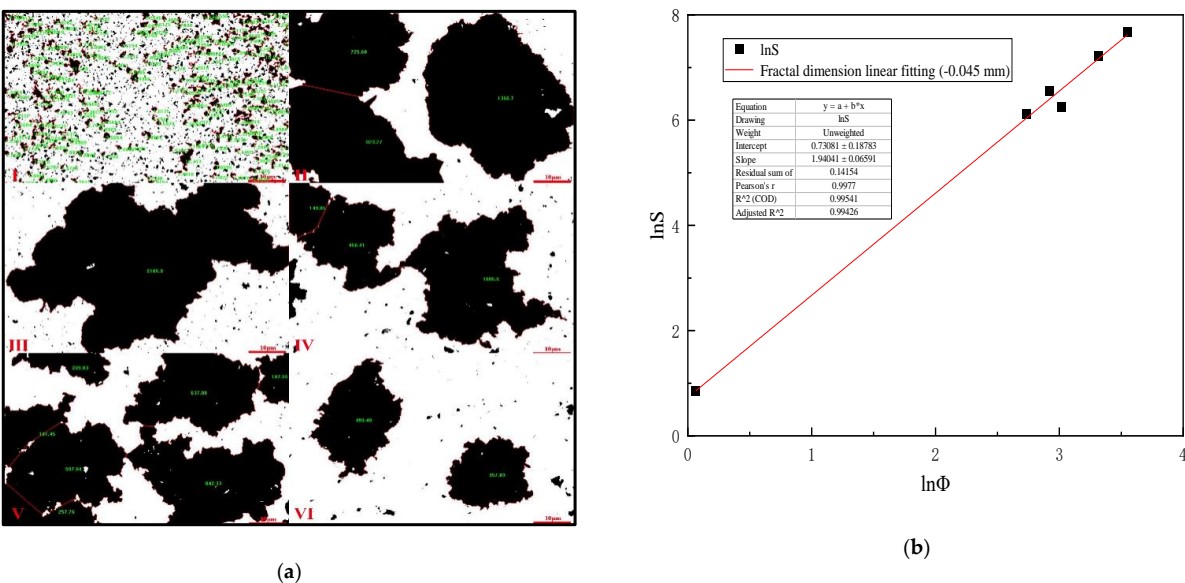

(**a**)　　　(**b**)

**Figure 11.** (**a**) Area diagram of −0.045 mm coal slime floc at different mixing speeds. (**I**) 0, (**II**) 200 r/min, (**III**) 400 r/min, (**IV**) 600 r/min, (**V**) 800 r/min, (**VI**) 1000 r/min; (**b**) Fractal dimension fitting diagram of settling floc of −0.045 mm coal slime.

*4.7. Results and Discussion*

(1)　When the mixing speed $N^* = 600$ r/min (impeller linear velocity $v^* = 0.63$ m/s), the equivalent diameter of settling floc of $-0.5$ mm slime is the largest; When $N^* = 200$ r/min or 400 r/min ($v^* = 0.21$ m/s or 0.42 m/s), the equivalent diameters of 0.5–0.25 mm, 0.25–0.125 mm, 0.125–0.075 mm, 0.75–0.045 mm and $-0.045$ mm slime settling floc are the largest.

(2)　The equivalent diameter is positively correlated with the size of slime, and the equivalent diameter of $-0.045$ mm slime settling floc is the largest; The size of slime is the dominant factor of fractal dimension change, and the fractal dimension of $-0.5$ mm slime settling floc is the largest.

(3)　When the mixing speed is 0–1000 r/min, the characteristic parameters are positively correlated with: statistical diameter and floc area; The negative correlations are as follows: particle size and floc area, floc perimeter, and statistical diameter.

**5. Flocculant Dissolution—Stirring Time Derivation Method**

*5.1. Derivation Method and Purpose*

According to the relationship of dissolution parameters of APAM with different molecular weights and the change of geometrical parameters of industrial mixing drum, a method of derivation of stirring time of dissolution of industrial flocculants was proposed. Stirring time and stirring power of stirring barrels with different geometric parameters could still be obtained by using the derivation method of stirring time.

*5.2. Application Formula of Stirring Time Method*

Table 11 shows that through the numerical fitting of Reynolds number and power number in stirring dissolution tests of APAM with different molecular weights, it is found that they are power correlated. The absolute value of the power index increases first and then decreases with the molecular weight, and reaches the peak when the molecular weight is 18 million, and the $R^2$ is the largest and the fitting error is the smallest. Therefore, APAM of 18 million molecular weight is recommended as the flocculating agent for dissolution, and the general formula of *Re-K* relationship applicable to each molecular weight is proposed: $K = 10Re^{-0.5}$. Similarly, through the numerical fitting of stirring power per unit volume and stirring time, it is found that they are linearly correlated, and the fitting slope and intercept are in a fluctuating trend. A general formula for the relationship between $P_v$ and $T$ applicable to each molecular weight is put forward: $T = -1.4P_v + 69.4$.

**Table 11.** Application formula fitting of stirring time method.

| Molecular Weight /(Million) | *Re* versus *K* Power Fitting Relationship | $P_v$ versus *T* Linear Fitting Relationship |
|---|---|---|
| 3 | $K = 9.3797Re^{-0.487}$ | $T = -1.5359P_v + 68.71$ |
| 6 | $K = 9.8157Re^{-0.512}$ | $T = -1.0376P_v + 66.148$ |
| 8 | $K = 9.6223Re^{-0.5}$ | $T = -1.1622P_v + 66.272$ |
| 10 | $K = 10.018Re^{-0.521}$ | $T = -1.5059P_v + 69.669$ |
| 12 | $K = 10.3Re^{-0.535}$ | $T = -1.6264P_v + 68.53$ |
| 16 | $K = 10.397Re^{-0.544}$ | $T = -1.4681P_v + 70.688$ |
| 18 | $K = 10.486Re^{-0.55}$ | $T = -1.3668P_v + 69.917$ |
| 20 | $K = 10.122Re^{-0.534}$ | $T = -1.5372P_v + 70.586$ |
| 22 | $K = 9.952Re^{-0.526}$ | $T = -1.6101P_v + 71.961$ |
| 25 | $K = 10.048Re^{-0.532}$ | $T = -1.1669P_v + 71.84$ |
| Average Fitting General Formula | $K = 10Re^{-0.5}$ | $T = -1.4P_v + 69.4$ |

*5.3. Contents of Derivation*

Figure 12 shows the flow of flocculant solution—stirring time derivation method Based on the law of APAM stirring dissolution, the viscosity $\mu$ and density $\rho$ of the hydrolyzed

solution are estimated according to the molecular weight of the selected flocculant, and the kinematic viscosity $\gamma$ of the solution is calculated. The impeller linear velocity $v$ is obtained according to the impeller diameter $d$ and stirring speed $N$. The Reynolds number $Re$ of the solution is calculated according to the kinematic viscosity $\gamma$, linear velocity $v$ and impeller diameter $d$. The power number $K$ is obtained by fitting the relation between the corresponding molecular weight $Re$ and $K$. The stirring power $P$ is calculated by the improved stirring formula $P = K\rho N^3 D^4 \left(\frac{d}{D}\right)^{5.45}$, which is suitable for APAM dissolution. The solution volume ratio is the ratio of the volume of the solution $V$ in the industrial stirring barrel and the volume of the glass constant temperature water bath test solution 0.01 m$^3$. By multiplying $P$ and $(V/0.01)$, the stirring power per unit volume $P_v$ is obtained. Finally, the stirring time $T$ is obtained according to the fitting relation between the corresponding molecular weight $P_v$ and $T$.

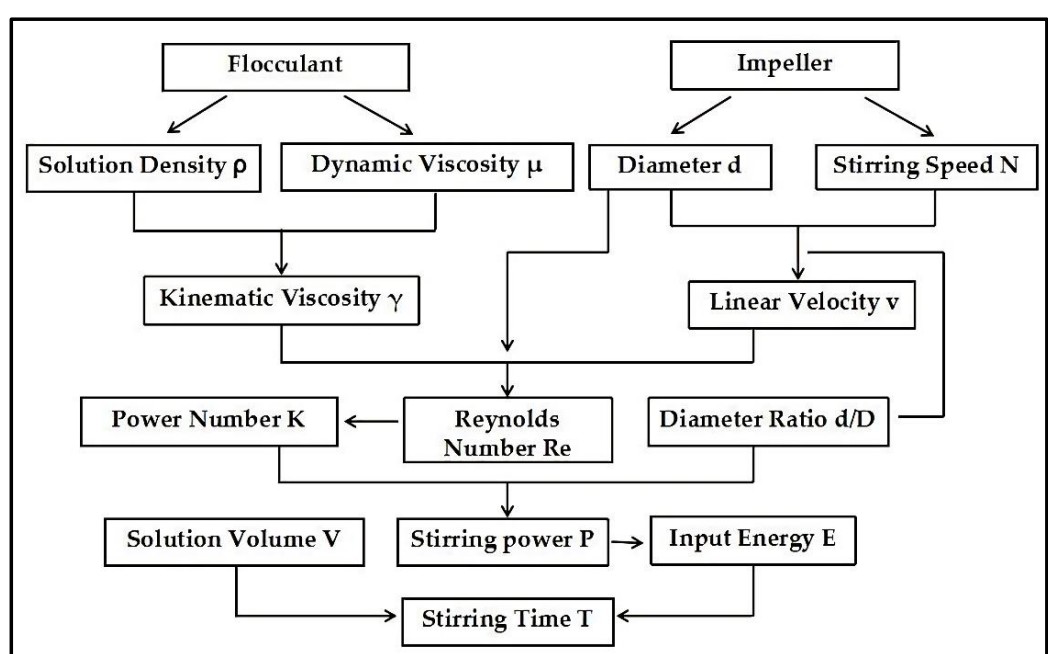

**Figure 12.** Flocculant dissolution—stirring time derivation method.

### 5.4. Industrial Application of Stirring Time Derivation Method

Table 12 shows the geometric parameters of industrial mixing drum, taking 18 million APAM as an example. The impeller linear velocity $v$ = 12.1 m/s was obtained based on the parameters $N$ = 1440 r/min, impeller Angle $\alpha$ = 36° and $d$ = 0.16 m. According to Table 2, the kinematic viscosity $\gamma$ = 2.3 × 10$^{-3}$ was taken and $Re$ = 841.7 was obtained according to the Reynolds number formula. According to the fitting formula of $K$ and $Re$ in Table 10, $K = 10Re^{-0.5}$, $K$ = 0.34; Using the improved stirring formula and solution volume ratio of 20:1, the unit volume of stirring power $P_v$ was calculated to be 7.56 kW/m$^3$. Finally, according to the fitting formula $T = -1.4P_v + 69.4$, the stirring time $T$ = 58.8 min was calculated.

There must be some errors between the derived stirring time method and the practice. The key is how to reduce the errors. The stirring time of flocculants is mainly affected by the viscosity of flocculants and the stirring speed. The viscosity ranges of the same flocculants are not very different, so the errors of Reynolds number, power criterion and stirring power are within a certain range, while the stirring speed is generally fixed. Therefore, the derivation method of stirring time has certain reference significance.

**Table 12.** Main component parameters of D680 mixing drum.

| Core Components | Diameter/cm | Length, Width, Height/cm | Note |
|---|---|---|---|
| Impeller | 16 | $10 \times 8 \times 6$ | Number 3, Angle 36° |
| Mixing drum | 68 | $68 \times 68 \times 68$ | Treatment 0.2 m³ |
| Stirring rod | 10 | $10 \times 6 \times 64$ | Connecting impeller |
| Motor | 26 | High 30 | Fixed speed 1440 r/min |
| Reduction gear | 26 | High 23 | Increase output torque |
| Coupling | Upper 26, Lower 30 | High 25 | Cushioning and damping |

## 6. Conclusions

(1) The optimum stirring speed is 800 r/min for 3–6 million molecular weight APAM corresponds to an impeller linear velocity is 0.84 m/s; The optimum stirring speed is 600 r/min for 8–16 million molecular weight APAM corresponds to an impeller linear velocity is 0.63 m/s; The optimum stirring speed is 800 r/min for 18–25 million molecular weight APAM corresponds to an impeller linear velocity is 0.84 m/s. Through data fitting, the improved stirring power formula for APAM dissolution is $P = K\rho N^3 D^4 \left(\frac{d}{D}\right)^{5.45}$.

(2) When the stirring speed is 200 to 1600 r/min, the corresponding impeller linear velocity is 0.21 to 1.68 m/s, the positively correlated dissolution parameters are kinematic viscosity and molecular weight, Reynolds number and linear velocity, power number and molecular weight, stirring power and linear velocity, stirring power and molecular weight, stirring time and molecular weight; There are negative correlations between Reynolds number and molecular weight, power number and linear velocity, stirring time and linear velocity.

(3) When the mixing speed is 600 r/min, the equivalent diameter of −0.5 mm slime settling floc is the largest. When the mixing speed is 200 r/min or 400 r/min, the equivalent diameters of 0.5–0.25 mm, 0.25–0.125 mm, 0.125–0.075 mm, 0.075–0.045 mm and −0.045 mm slime settling floc are the largest. The equivalent diameter is positively correlated with the size of slime, and the equivalent diameter of −0.045 mm slime settling floc is the largest. The size of slime is the dominant factor in the change of fractal dimension. The fractal dimension of −0.5 mm slime settling floc is the largest, and the fractal dimension is little affected by mixing speed.

(4) The general formula of the relation between Reynolds number *Re* and power number *K* is: $K = 10Re^{-0.5}$. The general formula of the relation between stirring power per unit volume $P_v$ and stirring time $T$ is: $T = -1.4P_v + 69.4$, and a flocculant dissolution—stirring time derivation method for different geometric parameters of agitator drum model or industrial agitator drum is proposed.

**Author Contributions:** Conceptualization, W.Z. and J.L.; methodology, J.L.; software, J.L.; validation, W.Z., C.C. and S.W.; formal analysis, J.L. and W.Z.; investigation, J.L., J.Z. and W.Z.; resources, W.Z.; data curation, J.L.; writing—original draft preparation, J.L.; writing—review and editing, J.L.; visualization, J.L.; supervision, W.Z.; project administration, W.Z.; funding acquisition, W.Z. All authors have read and agreed to the published version of the manuscript.

**Funding:** This research is supported by the Natural Sciences Foundation of Anhui Province (2108085ME160), the Open Foundation of State Key Laboratory of Mining Response and Disaster Prevention and Control in Deep Coal Mines (SKLMRDPC19KF11), the University Scientific Research Project of the Education Department of Anhui Province (KJ2021A0429), and the University Excellent Talents Training Funding Project of the Education Department of Anhui Province (gxyqZD2021109).

**Data Availability Statement:** The data that support the findings of this study are available from the corresponding author upon reasonable request.

**Acknowledgments:** Special thanks to Zhou Wei of Anhui University of Science and Technology for his guidance of this study, and also to all those who participated in this study.

**Conflicts of Interest:** The authors declare no conflict of interest. The funders had no role in the design of the study; in the collection, analyses, or interpretation of data; in the writing of the manuscript; or in the decision to publish the results.

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
