# Peer review of "Effect of Agitation on the Dissolution of APAM with Different Molecular Weights and the Equivalent Diameter of Coal Slime Settling Floc with Different Particle Sizes"

_minerals, doi:10.3390/min13020204_

Round 1

Reviewer 1 Report

The manuscript " Effect of Different Molecular Weights APAM on Stirring Dis- solution" has a relevant issue for the law of stirring dissolution of flocculants. However, there are several problems that make it impossible to publish the manuscript in the current format. I recommend the followings:

1.     Introduction: The difference between anionic and cationic flocculants is not elaborated here, so why does it lead directly to exploring the performance of anionic flocculants?

2.     Figure 1: Figure 1 does not look beautiful, it is recommended to beautify it.

3.     Line 119: Please explain in detail how the degree of clarification is determined.

4.     Line 177-179: Please explain in detail where is a certain stirring (linear velocity) speed range.

5.     Line 186-187, 203-204: Same as 4.

6.     Figure 6: It is suggested that the annotations in Figure 6 be enlarged and placed where they can be prominently seen.

7.     Line 250: Revise this sentence.

8.     Conclusion 2: The positioning of the rate of stirring speed is too vague.

9.     In line 140 and 143, Does K represent the same meaning?

10.  The obtained model has not been verified.

11.  Whether the laboratory size is affected by other factors when it is directly and equivalently magnified to industry?

Author Response

Response to Reviewer 1 Comments

Reviewer 2 Report

This paper aims to study the effect of different molecular weights APAM on stirring dissolution. Through APAM stirring dissolution test and APAM flocculation performance measurement test, explore the dissolution law of APAM with different molecular weight under different stirring speed. Then, a derivation method of stirring time for the dissolution of flocculant is put forward. Given the good idea and logicality, I suggest it could be published on Minerals with minor revision. The following is some comments.

1.    Some suggestions on the writing. 1) The conclusion of grammatical tenses is in the present tense, and the past tense can be used for experiments that have taken place. 2Full text academic terms to achieve complete unity.

2.    The title of 2.1, looks like it should be “Instruments and test system” based on the content. In addition, 2-1(3) experimental materials subject is not complete, should be improved.

3.    The significance of experiments and their direct logical connection should be made clear.

4.    How is the picture in the paper obtained and should be supplemented with a brief introduction.

5.    What is the source of the headings in Chapter 3.1, 3.2 and 3.3, and what is the meaning represented by the pictures in the whole article? It should be brief and concise.

6.    The origin and assignment of Table 5 in Chapter 4 should be supplemented.

7.    A meaning work required good references. Add couples of recent references.

Author Response

Response to Reviewer 2 Comments

Reviewer 3 Report

1. APAM should be mentioned in full right from the abstract instead of using abbreviation. It was only mentioned in full in page 2 line 72.

2. Anionic should not use capital letter, refer page 2 line 81.

3. Methods written in active sentences, refer page 3 line 99-103 and line 102-112

4. Need to check on x-axis label for Figure 2, 3 and 4 (sriring speed)

5. Numbering sequence for Figure 4, 5 and 6 need to be corrected, should start with (a) for each figure, not continuous.

6. Lack of citation of other researchers previous works in Results section to support authors' current results.

Author Response

Response to Reviewer 3 Comments

Reviewer 4 Report

The manuscript is on a topic of interest to me, and I think its results add something, but anyone reading it would almost assume no-one had published on the topic before. Thirteen of the fourteen references are all cited in the first paragraph of the Introduction – admittedly, it’s a long paragraph (too long), but these citations are just a rambling entry into aspects of flocculation, and not the subject of flocculant dissolution. Any proper scientific study needs to do a better job of acknowledging previous work, of which the following are some easily accessible examples:

Waters (1985) https://doi.org/10.1080/07349348508905151

Owen et al. (2002) https://doi.org/10.1016/S0301-7516(02)00035-2

Owen et al. (2007) https://doi.org/10.1016/j.minpro.2007.05.003

Garmsiri and Shirazi (2014) https://doi.org/10.1016/j.mineng.2014.05.011

The submitted work certainly goes beyond the above in several ways, such as by trying to quantify the impact of product molecular weight and in seeking to come up with some guidelines for larger-scale make-up. Where the previously published work is superior is in the actual flocculation results provided. Despite lines 115-120 implying a lot of flocculation tests were done, we are only shown Figure 5f, which is for one flocculant at an unspecified dosage. We don’t get to see the dosage response, which would be interesting as it would show the true impact of flocculation and the effectiveness of the make-up.

While the mineralogical composition of the solid sample is indicated, all we are told about the particle size is that its -0.5 mm. That is inadequate characterisation. As a minimum, there should be full sieve analysis to show at least the % under 45 micron, but a full laser particle size distribution would be more appropriate. If the sample is low in fines, it would certainly make me doubt the fractal dimensions that have been derived.

The authors claim the optimal stirring times are 50 to 70 minutes, but we are given no result to show that. This is important, as the stated times seem short to me, which indicates the possibility that the intensity of the mixing during make-up may be too high, leading to shear degradation at longer times – this is also dealt with extensively in the published literature, but gets no mention in this manuscript. In fact, I don’t believe there is any literature cited at all after Section 2 (i.e. no citations in Sections 3 and 4, which is where all the results and discussion are), and that almost never occurs in a quality journal publication.

My recommendation is that the authors be asked to conduct a major review to better explain the above concerns and properly cite the relevant literature, hopefully raising this to the standard expected of a journal publication. At present its more like a technical report.

Some other minor points (I could make more suggestions on the written English, but there’s no point going into that until the bigger issues are resolved):

·       I have never seen before multi-image figures be labelled continuously, such that Figure 4 has c and d as the labels, even though only two graphs are shown. Unless this is an MDPI peculiarity, each new figure should start at “a”.

·       The acronyms APAM and CPAM are used without being defined.

·       Figure 2 and 3 axis labels – “stirring” is spelt wrong.

·       Presumably Figure 5f is using the 18 million MW product, but it’s not stated in the caption.

Author Response

Response to Reviewer 4 Comments

Round 2

Reviewer 4 Report

The manuscript has been significantly improved. I still think the flocculation work is lacking, and I'd normally rather see dosages as grams of polymer per tonne of solid (the authors instead only give a volume dose of polymer), but with so few flocculation conditions studied, that's a minor point. I think its publishable now if the following small points are addressed.

The term APAM needs to be defined the first time it appears in the main text (Introduction). The authors currently do this in line 68, but APAM is first used in line 38.

Line 38: I recommend changing “agglomeration” to “aggregation”.

Line 41: “Appropriate dosage of APAM has a positive effect on kaolinite [2,3]” is a poor sentence. A positive effect in what way? Would read better as “Applying an appropriate dosage of APAM has a positive effect on kaolinite aggregation and subsequent settling [2,3].”

Line 46: I have no idea what “exhibit significant reflux capacity” means. Do the authors mean they can exhibit significant capacity to reform?

Line 55: change “dose” to “dosage”.

References: shouldn’t “Doi” be “DOI”?

Author Response

Response to Reviewer 4 Comments-2
